# How Well Do Text Embedding Models Understand Syntax?

**Yan Zhang** [*1]     **Zhaopeng Feng**[*2]     **Zhiyang Teng** [3]     **Zuozhu Liu**[2,4†]    **Haizhou Li**[1,5]

[1] National University of Singapore    [2]Zhejiang University [3]Nanyang Technological University
[4]Angelalign Inc., China [5]The Chinese University of Hong Kong, Shenzhen, China
chihyangteng@gmail.com
{zhaopeng.23,zuozhuliu}@intl.zju.edu.cn
{haizhou.li,eleyanz}@nus.edu.sg

## Abstract

Text embedding models have significantly contributed to advancements in natural language processing by adeptly capturing semantic properties of textual data. However, the ability of these models to generalize across a wide range of syntactic contexts remains under-explored. In this paper, we first develop an evaluation set, named **SR**, to scrutinize the capability for syntax understanding of text embedding models from two crucial syntactic aspects: **S**tructural heuristics, and **R**elational understanding among concepts, as revealed by the performance gaps in previous studies. Our findings reveal that existing text embedding models have not sufficiently addressed these syntactic understanding challenges, and such ineffectiveness becomes even more apparent when evaluated against existing benchmark datasets. Furthermore, we conduct rigorous analysis to unearth factors that lead to such limitations and examine why previous evaluations fail to detect such ineffectiveness. Lastly, we propose strategies to augment the generalization ability of text embedding models in diverse syntactic scenarios. This study serves to highlight the hurdles associated with syntactic generalization and provides pragmatic guidance for boosting model performance across varied syntactic contexts.

## 1 Introduction

Text embedding plays a significant role in a variety of natural language processing applications including language understanding (Du et al., 2020), information retrieval (Thakur et al., 2021) and question answering (Ni et al., 2021). Recently, many methods (Reimers and Gurevych, 2019; Gao et al., 2021; Ni et al., 2021; Neelakantan et al., 2022; Su et al., 2023), targeting at converting textual data into vector representations, have demonstrated a remarkable performance across massive text embedding benchmarks (Muennighoff et al., 2023).

---

[*] Equally Contributed.
[†] Corresponding author.

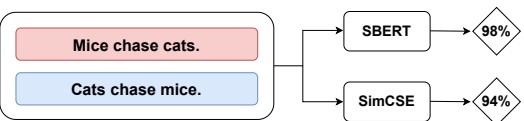

Figure 1: A probing example by using SBERT (all-MiniLM-L6-v2) and SimCSE (Roberta-Large) to compute the semantic similarity between "Cats chase mice" and "Mice chase cats". Both models exihibit high similarity socres at this simple task.

However, as we delve deeper into different hierarchies of language comprehension, the question that naturally arises is: *How well do these text embedding models understand syntax*? For example, could the state-of-the-art text embedding models understand and distinguish the difference between "Cats chase mice" and "Mice chase cats"? Syntax, constituted by a set of rules that define sentence structures, forms a pivotal aspect of natural language. It integrates both heuristics and compositional elements, establishing the bedrock for the expansive and intricate nature of human language (Manning and Schütze, 1999; Jurafsky and Martin, 2000). A thorough comprehension of syntax is essential for a text embedding model to effectively ascertain the relationships among words, thereby facilitating a level of language understanding that mirrors human cognitive processes. Moreover, as text embedding models are increasingly deployed in LLM-based agents and real-world applications, ensuring that they maintain a solid understanding of syntax is critical to guarantee their reliability and efficacy.

In this study, we address this question by introducing a new evaluation set called SR designated to probe the ability of text embedding models from three syntactic aspects (Partee, 1995; Gibson, 1998; Gildea and Jurafsky, 2000; Mitchell and Lapata, 2010; McCoy et al., 2019; Linzen and Baroni, 2020): 1) **S**tructural heuristics: the rules and patterns that govern sentence structures, and 2) **R**elational understanding among concepts: the

models' capability to infer relationships between different concepts in text. These dimensions represent the multifaceted nature of syntax.

We take this inquiry a step further by conducting an incisive analysis to uncover the underpinnings of these limitations. By examining various models and their responses to syntactic challenges, we delineate the factors that contribute to the manifestation of these limitations and the reasons why they have eluded detection in conventional benchmarks. Recognizing these challenges is a precursor to addressing them. Finally, we show that simply augmenting the training Data with SR-like examples, which can be generated through ChatGPT with designed prompts, can significantly enhance the generalization capabilities of text embedding models in syntactically diverse settings.

This research is aimed at shedding light on the challenges of syntactic generalization in text embedding models. By establishing a more rigorous evaluation set and proposing strategies for enhancement, we contribute to the advancement of text embedding models that are capable of nuanced understanding and high performance across varied syntactic contexts. Our work provides valuable guidance and sets the stage for future research aimed at achieving more syntactically aware and robust text embedding models. SR benchmark and code are released [1].

## 2  SR Benchmark

Typical evaluation sets have often been inadequate in rigorously assessing a model's understanding of syntax. They tend to focus on high-level performance metrics, while overlooking the finer aspects of syntactic understanding. SR was designed to address the limitations and specifically evaluate text embedding models from two important syntactic aspects: Structural heuristics, and Relational understanding. These dimensions were chosen due to their fundamental roles in natural language understanding. In this section, we introduce the construction process of the SR benchmark.

### 2.1  Foundation Corpus for SR Construction

For constructing the SR benchmark, it was vital to base it on rich and diverse foundational corpus that encompass different domains and compositional structures. These corpus include:

---

[1] SR Benchmark, code, prompts, and other details are publicly available at https://github.com/fzp0424/SR.

**STS:** We adopt the STS benchmark (Cer et al., 2017), which comprises a selection of STS tasks organized in the context of SemEval between 2012 and 2017. The dataset include 1,379 sentence pairs from image captions, news headlines and user forums. They provide a range of sentences with varying complexity and structures, making them an ideal starting point for our SR benchmark.

**SICK:** SICK(Sentences Involving Compositional Knowledge) (Marelli et al., 2014) includes 9927 sentence pairs that are rich in the lexical and syntactic phenomena. They provide a range of sentences with diverse compositional structure for our SR benchmark.

**CQADupStack:** CQADupStack (Hoogeveen et al., 2015) is a dataset derived from Stack Exchange and is geared towards the identification of duplicate questions in community Question-Answering forums. It brings in diversity from real-world queries and their paraphrased versions.

**Twitter:** The Twitter dataset (Xu et al., 2015) consists of pairs of tweets together with a crowd-annotated score if the pair is a paraphrase. Including this dataset allows the SR benchmark to account for the informal and concise nature of social media text, which often involves unique linguistic constructs.

**BIOSSES:** BIOSSES (Soğancıoğlu et al., 2017) is a dataset designed to evaluate biomedical semantic similarity with 100 testing pairs. By including BIOSSES, the SR benchmark encompasses the domain-specific language used in biomedical texts.

**AskUbuntu:** AskUbuntu (Lei et al., 2015) is a collection of user posts from the technical forum AskUbuntu. The inclusion of AskUbuntu allows for the SR benchmark to encompass complex and technical questions typically encountered in community-driven platforms.

By utilizing these datasets, the SR benchmark is well-equipped to evaluate text embedding models' syntactic understanding across varying domains and compositional structures. More important, the recent text embedding models have already demonstrated remarkable performance on these benchmarks. This configuration not only ensures a holistic evaluation but also effectively pinpoints the models' strengths and weaknesses in diverse settings.

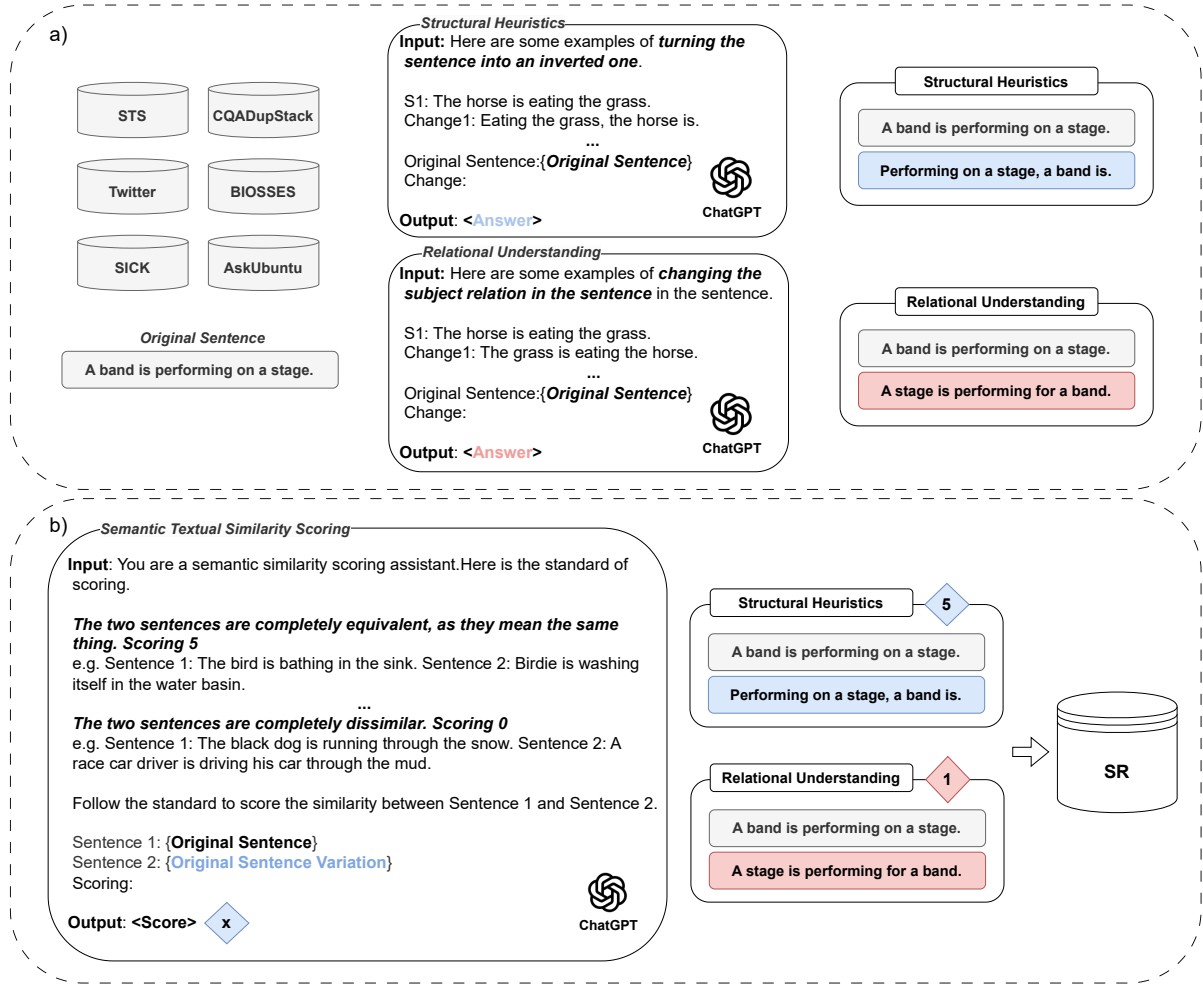

Figure 2: Workflow of SR benchmark construction. The workflow begins with the selection of foundation datasets, followed by the generation of data probing structural heuristics and relational understanding among concepts. Subsequently, the generated data is annotated with a score to represent the semantic similarity of sentence pairs. Finally, the assembled SR benchmark is utilized to evaluate the syntactic understanding capabilities of text embedding models.

## 2.2 Assessing Structural Heuristics

To assess how well text embedding models comprehend structural heuristics, we focus on the transformation of sentences into different syntactic structures (Gibson, 1998). In particular, we emphasize the exchange between active and passive voice as well as creating inverted sentences or partially inverted sentences.

**Active and Passive Voice Exchange:** In the active voice, the subject of the sentence performs the action, whereas in the passive voice, the subject is acted upon. We transform sentences from the foundation dataset into their passive or active counterparts. For instance, an active sentence such as "The chef cooked the meal" can be transformed into its passive equivalent, "The meal was cooked by the chef." Text embedding models should rec-

ognize that these sentences convey the same action but with a different focus and structure.

**Inverted and Partially Inverted Sentences:** Inverted sentences involve reversing the canonical subject-verb-object order, often for emphasis or stylistic purposes. Partial inversion involves only a segment of the sentence being inverted. Assessing text embedding models on their ability to understand these structural changes is critical for gauging how well they can adapt to varied syntactic constructs. For instance, a standard sentence like "The team has won the championship." can be converted into an inverted sentence, such as "Won the championship has the team.", while a partially inverted example might be, "The championship, the team has won.". The models should be able to understand that, despite the alteration in structure, the

core information remains the same.

## 2.3 Assessing Relational Understanding

The arrangement and relationship of concepts within a sentence contribute significantly to the meaning and interpretation of the text. Changing the order or relationship of these concepts may radically alter the sentence's meaning.

**Concept Order Manipulation:** Concept order within sentences often plays a critical role in conveying meaning. We manipulate the order of concepts in sentences from the foundation dataset to create new sentences. The objective is to examine whether text embedding models can recognize how these manipulations impact the meaning of sentences. For example, consider the sentence: "Tom likes Taylor Swift.". By altering the order of the concepts, we get: "Taylor Swift likes Tom.". Though structurally similar, these sentences have completely different meanings. The models should recognize the shift in the subject and object and the corresponding change in the meaning.

**Conceptual Relationships Perturbations:** Furthermore, we evaluate the models' ability to understand various relationships among concepts, such as cause-effect, part-whole, and synonymy etc. For instance, in a sentence like "Rain causes floods.", the model should identify the cause-effect relationship between "rain" and "floods". As another example, consider the sentence "He read the book because he was interested in history.". By changing the order, e.g., "Because he was interested in history, he read the book.", the meaning remains the same, but the structure has changed. The model should be able to recognize the constancy in the meaning despite the alteration in syntax.

| | STS | | |
| --- | --- | --- | --- |
| | Original Samples | Modified Samples | Modification Rate |
| Structural Heuristics | 400 | 3 | 0.75% |
| Relational Understanding | 400 | 4 | 1.00% |

Table 1: Statistical breakdown of the revision rate for modified sentences in STS.

| | Twitter | | |
| --- | --- | --- | --- |
| | Original Samples | Modified Samples | Modification Rate |
| Structural Heuristics | 400 | 5 | 1.25% |
| Relational Understanding | 400 | 8 | 2.00% |

Table 2: Statistical breakdown of the revision rate for modified sentences in Twitter.

| | CQADupStack | | |
| --- | --- | --- | --- |
| | Original Samples | Modified Samples | Modification Rate |
| Structural Heuristics | 400 | 9 | 2.25% |
| Relational Understanding | 400 | 6 | 1.50% |

Table 3: Statistical breakdown of the revision rate for modified sentences in CQADupStack.

## 2.4 Annotation and Validation

As presented in Figure 2, we first utilize ChatGPT to generate sentence pairs that probes the above three syntactic dimensions with specific designed prompts. Afterwards, ChatGPT is also utilized to annotate the sentence pairs with semantic similarity scores. Through a process of duplication and filtering of sentences that are too short, we manage to assemble a dataset comprising 9,424 sentence pairs for each syntactic dimension in the SR benchmark.

To verify the generation and annotation quality, we first conduct additional validation by randomly sampling 800 sentence pairs from the SR benchmark. Any sentences that are inaccurately generated will be corrected as needed. Any sentences that are inaccurately generated will be modified as needed. Table 1, Table 2, and Table 3 present the statistical breakdown of the revision rate for these corrected sentences. Moreover, we utilize ChatGPT to assign similarity scores to the standard STS-B evaluation set, and we compute the correlation between these scores and the STS-B standard annotated scores.

Upon completion of these exercises, we find that the revision rate is no more than 3% and the correlation scores between ChatGPT annotations and human annotations stand at 83.8%. This indicates that ChatGPT is proficient in generating syntax varied sentences and annotations that closely resonate with human evaluations, thereby affirming the reliability and authenticity of the SR benchmark.

## 3 Evaluating Test Embedding Models on SR

In this section, we employ the SR Benchmark to evaluate the syntactic understanding capabilities of five state-of-the-art text embedding models, including SentenceBERT (Reimers and Gurevych, 2019), SimCSE (Gao et al., 2021), Sentence-T5 (Ni et al., 2021), One embedder (Su et al., 2023) and OpenAI Embedding (Neelakantan et al., 2022). Detailed information on these models can be found in their original papers. Following previous works, we employ Spearman's correlation as the evaluation

| Model | STS | SICK | AskUbuntu | CQADupStack | Twitter | BIOSSES | Avg |
|---|---|---|---|---|---|---|---|
| *Structural Heuristics* | | | | | | | |
| SimCSE-BERT-Base-sup | 19.14 | 21.82 | 35.38 | 14.17 | 19.68 | 37.27 | 24.73 |
| SimCSE-BERT-Large-sup | 19.93 | 23.52 | 33.27 | 15.60 | 18.59 | 35.13 | 24.34 |
| SimCSE-RoBERTa-Large-sup | 22.97 | 26.88 | 37.87 | 19.20 | 15.88 | 35.42 | 26.37 |
| SBERT-all-MiniLM-L6-v2 | 13.08 | 14.83 | 36.56 | 19.69 | 10.99 | 13.17 | 18.05 |
| SBERT-all-mpnet-base-v2 | 18.35 | 24.62 | 37.62 | 24.39 | 13.14 | 16.00 | 22.35 |
| Sentence-T5-Base | 15.60 | 21.55 | 35.34 | 23.31 | 20.80 | 35.45 | 25.34 |
| Sentence-T5-Large | 17.03 | 22.83 | 36.20 | 24.30 | 21.83 | 35.90 | 26.35 |
| Sentence-T5-XL | 17.86 | 23.25 | 37.02 | 26.93 | 21.24 | 35.79 | 27.01 |
| Instructor-Base | 17.48 | 18.73 | 33.94 | 21.06 | 15.95 | 23.31 | 21.75 |
| Instructor-Large | 20.96 | 26.35 | 36.36 | 22.09 | 19.34 | 29.79 | 25.81 |
| Instructor-XL | 20.46 | 18.32 | 34.89 | 19.97 | 20.28 | 26.55 | 23.41 |
| OpenAI (text-embedding-ada-002) | 15.35 | 21.55 | 36.25 | 26.50 | 17.90 | 22.40 | 23.32 |
| *Relational Understanding* | | | | | | | |
| SimCSE-BERT-Base-sup | 30.39 | 38.82 | 44.87 | 30.15 | 38.07 | 41.79 | 37.35 |
| SimCSE-BERT-Large-sup | 38.8 | 43.48 | 44.13 | 30.64 | 40.09 | 48.80 | 41.16 |
| SimCSE-RoBERTa-Large-sup | 48.34 | 48.16 | 46.51 | 31.28 | 40.22 | 50.43 | 44.16 |
| SBERT-all-MiniLM-L6-v2 | -2.72 | 18.30 | 47.53 | 25.14 | 25.70 | 11.69 | 20.94 |
| SBERT-all-mpnet-base-v2 | 30.39 | 40.42 | 50.54 | 30.29 | 26.09 | 0.26 | 29.67 |
| Sentence-T5-Base | 33.61 | 39.36 | 52.59 | 35.72 | 42.53 | 47.63 | 41.91 |
| Sentence-T5-Large | 51.08 | 50.20 | 51.09 | 39.85 | 44.80 | 53.82 | 48.47 |
| Sentence-T5-XL | 58.64 | 53.34 | 55.12 | 42.10 | 48.21 | 53.41 | 51.80 |
| Instructor-Base | 1.33 | 28.06 | 48.64 | 31.46 | 36.75 | 18.68 | 27.49 |
| Instructor-Large | 21.88 | 40.88 | 49.41 | 38.50 | 40.71 | 41.74 | 38.84 |
| Instructor-XL | 28.97 | 37.67 | 48.94 | 39.30 | 42.34 | 35.24 | 38.74 |
| OpenAI (text-embedding-ada-002) | 19.10 | 33.20 | 48.70 | 24.60 | 33.20 | 29.90 | 31.45 |

Table 4: Results of five text embedding models on the SR Benchmark. Spearman's correlation is reported.

metric to assess how well the relationship between the cosine similarities of the sentence pairs and the annotated scores.

We present the evaluation results on the SR benchmark in Table 4. Though these models have previously exhibited remarkable performance on foundational test datasets, they all perform poorly with low correlations on the SR benchmark. This suggests that existing text embedding models have not been optimized sufficiently to address the syntactic understanding challenge.

Interestingly, we observe that, despite being trained solely on natural language inference datasets, SimCSE outperforms SBERT, Instructor and OpenAI embedding models. These competing models often have more diverse supervised pairwise training datasets or a greater number of parameters, or both. For instance, SimCSE achieves performance with an average Spearman's correlation of 44.16% on the benchmark for assessing relational understanding, while SBERT, Instructor and OpenAI embedding models achieve an average Spearman's correlation of 29.67%, 38.74% and 31.45%, respectively. This could indicate that training on natural language inference tasks may

provide valuable syntactic understanding that isn't captured through mere parameter scale or breadth of training data. Sentence-T5, which is trained on a large scale of web-based question-answering datasets, demonstrates most robust performance. This might suggest that the diversity and complexity found in web-based question-answering data could be beneficial for embedding models in capturing syntactic aspects. We hypothesize that the web-based question-answering data, often encompassing a wide range of topics, styles, and structures, are likely to present a richer and more varied set of syntactic compositions compared to other data types.

We use tree kernel (Collins and Duffy, 2002; Yu and Sun, 2022) as a measure of sentence structure diversity. The central idea of tree kernel is to count the number of common subtrees between two constituency pars. We compare the syntactic diversity of two corpora: WebQA (233k sentences selected) and NLI (275k sentences selected). We use StanfordCoreNLP to obtain constituency parse trees of sentences. Then, we randomly select 1,000 parse trees and use the tree kernel to calculate their similarity. The results are shown in Table 5. The lower

| Corpus | Tree Kernel Similarity |
|--------|------------------------|
| WebQA  | 0.064                  |
| NLI    | 0.072                  |

Table 5: Comparison of tree kernel similarity between WebQA and NLI corpora.

the score, the more diverse the sentence structures are. As we can see, sentences in WebQA have lower tree kernel similarity than those in NLI, indicating that WebQA has more diverse sentence patterns and structures. This diversity could expose the model to a more comprehensive spectrum of syntactic elements, instilling a deeper sense of syntax into the embedding models. Since prevailing text embedding models like SBERT, SimCSE, etc. are mainly trained on NLI, we argue that Sentence-T5, which is trained on WebQA, can capture more syntactic nuances of sentences. Additionally, we found that these models generally struggle in the CQADupStack domain but show relatively better performance in the AskUbuntu domain. This discrepancy may be due to the specific challenges and complexities associated with the CQADupStack domain compared to the AskUbuntu domain.

Moreover, a noteworthy trend across all the text embedding models we assessed was the consistent pattern of comparatively better performance on the benchmark for relational understanding, as opposed to benchmarks for structural heuristics. This suggests that the current text embedding models may be more adept at capturing relations among concepts in sentences than they are at grappling with the finer points of syntax, such as sentence structures. This superior performance in capturing relations could be a result of the training data, which often tends to focus on semantic relationships. However, it might also indicate that the models have an innate aptitude for discerning semantic relations as compared to understanding complex syntactic structures.

## 4 Analysis

### 4.1 Shortcomings of Traditional Evaluation Paradigms

It is noteworthy that many text embedding models, despite exhibiting poor syntactic understanding, have consistently shown high performance on semantic embedding matching tasks. A primary reason for this is that these models tend to capture semantic content effectively, even when syn-

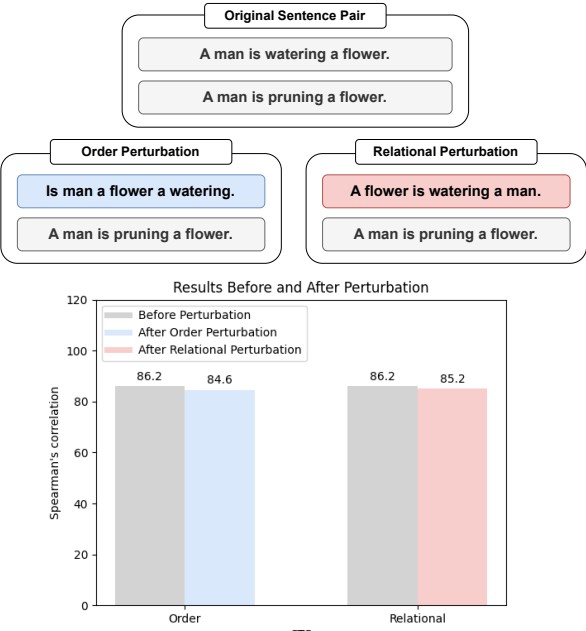

Figure 3: Retrain SBERT under syntactic perturbations such as randomized word order and exchanged relationships among concepts. Despite significant alterations in syntactic structure, the models continue to exhibit high similarity scores, highlighting their reliance on semantic content over syntactic understanding.

tactic information is not properly encoded. The rich semantic information contained in the large-scale corpora on which these models are trained allows them to make reasonably accurate predictions about the similarity between sentence pairs based on shared content words, irrespective of their syntactic structure.

Traditional evaluation paradigms in semantic matching tasks often lack sensitivity to syntactic nuances. Specifically, they tend to reward models for correctly identifying semantic content overlap, but do not penalize them adequately for failing to recognize distinct syntactic constructions. As such, a model might receive a high similarity score for two sentences that share similar content but have different syntactic structures, thus effectively ignoring syntax.

To empirically demonstrate that models can achieve high performance on semantic embedding matching tasks without proper syntactic understanding, we conducted an experiment where we manipulated the input data to the text embedding models to remove or alter syntactic information while preserving semantic content. The models' performance was then evaluated in terms of their ability to accurately measure semantic similarity in the manipulated data.

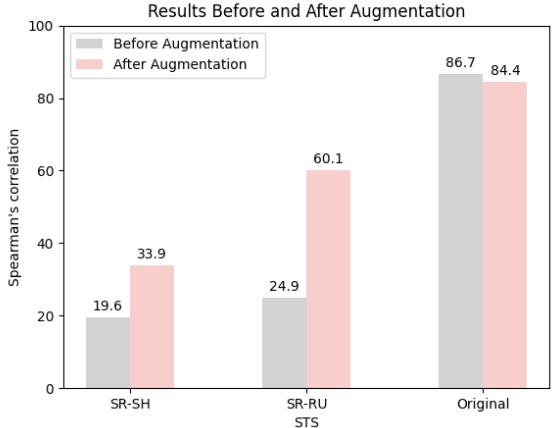

Figure 4: Enhancing syntactic proficiency of SBERT through strategic data augmentation.

As presented in Figure 3, we created two variants of perturbations on the STS dataset – one with randomized word order, and another with exchanged relationships among concepts – both of which significantly transform the syntactic architecture of the sentences. STS training dataset consists of sentence 1, sentence 2, and its semantic similarity score is marked by humans. We denote it as **[S1, S2, score]**. In this section, we fixed sentence 1 and did the perturbation on sentence 2. Each sentence pair is changed into **[S1, Perturbed S2, score]**. Remarkably, after retraining SBERT with perturbed input while keeping the original annotations unaltered, we discerned that the text embedding models continued to yield high similarity scores on the standard STS evaluation set. To illustrate, the correlation score following relational perturbation stood at 85.2%, nearly on par with the original score of 86.2%.

This experiment underscores the models' proclivity to latch onto semantic content, often overlooking syntactic structures when adjudicating the similarity between sentences. These findings prompt a more rigorous and discerning evaluation paradigm that factors in both semantic and syntactic elements, paving the way for more holistic and accurate assessments of text embedding models.

## 4.2 Simple Solution: Enhancing Syntactic Understanding through Targeted Data Augmentation

In light of the observation that contemporary text embedding models tend to exhibit weak syntactic understanding, one simple approach entails enriching the training data with examples that mirror

those in the SR benchmark, which is specifically designed to probe syntactic understanding. The process involves creating additional training examples that emulate the two facets of the SR benchmark: structural heuristics and relational understanding among concepts. Such new sentences can be synthesized by altering the original sentences through ChatGPT with designed prompts. Similar to Section 4.1, we fix sentence 1, do syntactic changes on each sentence 1 and ask ChatGPT(gpt-3.5-turbo-0301) to score the semantic similarity between the fixed sentence 1 and its perturbed one based on similarity scores with explanations and English examples from (Cer et al., 2017). We manually verified the generation reliability and annotation rationality. Each sentence pair can be written as **[S1, Perturbed S1, re-score]**.

For our experiment, we adopt SBERT as the base model and conducted experiments by enhancing the STS training dataset with 10,000 SR-like examples for each of the syntactic dimensions (structural heuristics and relational understanding among concepts). We then retrained SBERT on the augmented dataset. We choose microsoft/mpnet-base as our raw model which is also the basic model of the best SBERT text embedding model SBERT-all-mpnet-base-v2. We follow Sentence-BERT training settings (Reimers and Gurevych, 2019), use the regression objective function, a batch-size of 16, 4 training epochs, Adam optimizer with learning rate 2e-5, and a linear learning rate warm-up over 10% of the training data. Our default pooling strategy is MEAN. We save the best parameters according to the dev set at the end of each epoch. At evaluation time, we compute the cosine-similarity between the sentence embeddings.

Figure 4 presented the results. We can observe that the incorporation of the augmented training examples led to a substantial improvement in the performance on the SR evaluation set, especially in the aspect of relational understanding. Specifically, the Spearman's rank correlation coefficient in relational understanding witnessed a remarkable increase from 24.9% to 60.1%. This indicates that the augmented data effectively enabled the model to develop a better grasp of the syntactic relationships between concepts.

However, while the performance improvement on the SR evaluation set is significant, it is also important to observe how this augmentation impacts the performance on the original STS test set (STS-

B). We noticed a slight decline in the performance on the STS-B test set, with the score decreasing from 86.7% to 84.4% after data augmentation. This slight decrease suggests that while the model became more proficient in understanding complex syntactic structures, it may have marginally compromised on some aspects it had initially learned.

This experiment demonstrates the feasibility and potential effectiveness of targeted data augmentation as a means to enhance the syntactic understanding of text embedding models. It also emphasizes the importance of careful data curation and balanced training to ensure that improvements in one area do not come at the expense of performance in another. Future work can focus on refining this approach to optimize the trade-offs and achieve improved performance across different dimensions of syntactic understanding.

## 5 Related Work

**Text Embeddings** Text representation learning is a fundamental task in natural language processing. In recent literature, contrastive learning (Hjelm et al., 2019; He et al., 2020; Chen et al., 2020) has emerged as the dominant paradigm for training text embedding (Carlsson et al., 2020; Zhang et al., 2020; Gao et al., 2021; Yan et al., 2021; Ni et al., 2021; Chuang et al., 2022; Neelakantan et al., 2022; Su et al., 2023). These approachese typically involves the use of large pairwise pretraining datasets with rich compositional structures, wherein the model is optimized to distinguish between similar and dissimilar text samples. Contrastive pretraining is geared towards optimizing text embedding matching. As a result, most text embedding models (Reimers and Gurevych, 2019; Gao et al., 2021; Ni et al., 2021) are evaluated on the basis of their performance in semantic similarity matching tasks (Cer et al., 2017; Muennighoff et al., 2023). However, these tasks not sufficiently encompass the complexity of syntax in natural language, which involve not only the arrangement of words but also their relationships and compositional semantics.

**Syntax Probing in NLP** Syntactic Probing seeks to understand to what extent the learned representations in NLP models capture syntactic information and how such information can be effectively extracted and analyzed. There have been numerous studies focused on this topic, each offering unique insights and methods for syntactic generalization.

(Dyer et al., 2016; Linzen et al., 2016; Hupkes and Zuidema, 2017; Conneau et al., 2018; Lin et al., 2019; McCoy et al., 2019; Shen et al., 2020; Newman et al., 2021). For example, Conneau et al. (2018) assess the syntactic generalization of modern sentence encoders through a series of selected downstream tasks. Their approach is more geared towards understanding what properties are encoded in the sentence embeddings and how these embeddings can be used for various NLP tasks. On the other hand, McCoy et al. (2019) developed the HANS dataset to specifically examine the performance of Natural Language Inference (NLI) models, particularly focusing on whether these models are adopting superficial syntactic heuristics over a deeper semantic understanding. Their work is especially relevant in the context of NLI tasks and aims to discern the methods that models use to arrive at decisions.

In contrast, our work aims to construct a benchmark for directly evaluating the syntactic capabilities of text embedding models, without being confined to a specific task like NLI. The primary goal is to facilitate the selection of robust and effective text embedding models for training in various applications by providing a direct measure of their syntactic understanding. This benchmark thus serves as an essential tool for researchers and practitioners who are looking to employ text embedding models that are both syntactically sound and effective in real-world applications.

## 6 Conclusion

This paper highlights the shortcomings of current text embedding models in understanding syntax. We introduced the SR benchmark to analyze models across two syntactic dimensions: structural heuristics and relational understanding. Our findings reveal that while these models are adept at semantic tasks, they struggle with syntax. We proposed a data augmentation technique using examples tailored for syntactic understanding, leading to notable performance gains on the SR benchmark. However, a slight performance dip was observed on the original test set. Future work could refine augmentation strategies to balance syntactic and semantic learning and create more holistic evaluation benchmarks.

# 7 Limitation

One limitation is that the SR benchmark may not comprehensively cover all the intricacies of natural language syntax. Real-world text data can be vastly more complex and varied. Also, the enhancement in syntactic understanding was primarily based on simple data augmentations, while a augmentation strategy is on demand to ensure that enhancing syntactic understanding does not come at the expense of semantic comprehension.

## Acknowledgements

We would like to thank all the reviewers for their constructive comments. This research is supported by the Agency for Science, Technology and Research (A*STAR) under its AME Programmatic Funding Scheme (Project No. A18A2b0046); Zhiyang Teng is partially supported by CAAI-Huawei MindSpore Open Fund (CAAIXSJLJJ-2021-046A).

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

## A    Lexical Compositionality

We also do additional research which we call Lexical Compositionality. It refers to how the individual meanings of words come together to form the composite meaning of the larger linguistic structure in which they are embedded. For evaluating text embedding models' adeptness in lexical compositionality, we create variants of the sentences in the foundation datasets through the insertion or replacement of different types of words: *nouns*, *adjectives*, and *verbs*. The modified sentences retain the basic structure of the originals but feature an additional or changed word, either a noun, adjective, or verb. This method of dataset generation aims to examine whether text embedding models can adapt to and accurately capture the semantic changes brought by these insertions and replacements.

**Noun:** The insertion or replacement of a noun is anticipated to modify the subject matter or the entities referred to within the sentence. For instance, the original sentence, "The cat jumped over the fence," could be transformed into "The cat jumped over the garden fence," by inserting the noun "garden". Similarly, by replacing the noun "cat" with "dog", we rewrite the original sentence as "The dog jumped over the fence". This method allows us to gauge the text embedding models' ability to discern the introduction of new elements within the sentence's subject matter and evaluate their sensitivity and adaptability to these semantic alterations.

**Adjective:** The insertion or replacement of an adjective adds or changes a descriptive aspect to the sentence, and it is imperative for a text embedding model to recognize how these change the sentence's attributes or qualities. For instance, by modifying the sentence "The car is fast." to "The leading car is incredibly fast.", we can test the model's ability to comprehend the enhanced emphasis on the car's position and its speed. The robustness of embedding models can be verified.

**Verb:** The insertion or replacement of a verb can modify the actions or states conveyed in the sentence while the verb alter may radically transform its essence. Thus, it's crucial for text embedding models to detect how this influences the sentence's dynamics. For instance, changing "She reads books." to "She reads and enjoys books." by inserting the verb "enjoys" examines the model's capability to understand the introduction of an additional action. In a similar way, when the sentence "Two kids are walking on a path in the woods." is transformed into "Two kids are fighting on a path in the woods," through verb replacement, it serves to test the model's proficiency in redirecting its focus to the altered semantic components of the sentence.

We follow experiment settings in Section 3. The results are reported in Table 6.

## B    Generated Example

Table 7 showcases an example of sentence perturbation.

| Model | STS | SICK | AskUbuntu | CQADupStack | Twitter | BIOSSES | Avg |
|---|---|---|---|---|---|---|---|
| *Lexical Compositionality* | | | | | | | |
| SimCSE-BERT-Base-sup | 32.77 | 27.89 | 45.54 | 5.01 | 17.86 | 40.06 | 28.19 |
| SimCSE-BERT-Large-sup | 33.29 | 28.99 | 45.18 | 4.49 | 17.55 | 40.74 | 28.37 |
| SimCSE-RoBERTa-Large-sup | 38.24 | 35.32 | 48.06 | 6.59 | 19.64 | 42.69 | 31.76 |
| SBERT-all-MiniLM-L6-v2 | 25.00 | 19.21 | 46.93 | 6.88 | 17.78 | 25.13 | 23.49 |
| SBERT-all-mpnet-base-v2 | 30.82 | 28.96 | 45.53 | 7.92 | 18.14 | 28.11 | 26.58 |
| Sentence-T5-Base | 34.24 | 23.90 | 48.30 | 6.39 | 22.41 | 36.00 | 28.54 |
| Sentence-T5-Large | 36.11 | 24.48 | 48.38 | 6.61 | 21.61 | 38.79 | 29.33 |
| Sentence-T5-XL | 37.49 | 25.94 | 48.81 | 7.70 | 21.46 | 36.31 | 29.62 |
| Instructor-Base | 28.02 | 21.48 | 45.61 | 7.31 | 18.20 | 26.96 | 24.76 |
| Instructor-Large | 32.69 | 28.00 | 46.92 | 8.31 | 18.68 | 39.30 | 29.15 |
| Instructor-XL | 28.57 | 20.86 | 49.97 | 7.81 | 20.38 | 31.18 | 26.46 |
| OpenAI(text-embedding-ada-002) | 26.43 | 26.70 | 49.77 | 6.67 | 20.73 | 26.37 | 26.11 |

Table 6: Results of five text embedding models on the Lexical Compositionality Benchmark. Spearman's correlation is reported.

| *Input:* A group of people are nervous about crossing the water. | |
|---|---|
| *Structural Heuristics* | |
| **Prompt:** Change the active and passive voice of the sentence. Turn the sentence into an inverted one. | Score |
| Crossing the water makes a group of people nervous. | 4 |
| Nervous about crossing the water, a group of people are. | 4 |
| *Relational Understanding* | |
| **Prompt:** Change the subject relation in the sentence, it needs you to change your thinking habits and say something anticonventional. | Score |
| The water is nervous about crossing a group of people. | 1 |
| *Lexical Compositionality* | |
| **Prompt:** Randomly add or replace nouns / verbs / adjectives in the sentence, but keep the structure and other parts of the sentence unchanged. | Score |
| A group of tourists are nervous about crossing the water. | 4 |
| A group of people are nervous about traversing over the water. | 5 |
| A group of anxious people are afraid of crossing the deep water. | 3 |

Table 7: Example of generated perturbations given the input sentence and specific prompts.