# OpenReview forum: "How Well Do Text Embedding Models Understand Syntax?"
_EMNLP/2023/Conference — EMNLP 2023 Findings_

### Official Review · Reviewer_neEv · 2023-07-25

**Typos Grammar Style And Presentation Improvements:** The best results in the tables should…
**Soundness:** 3

**Excitement:**

3: Ambivalent: It has merits (e.g., it reports state-of-the-art results, the idea is nice), but there are key weaknesses (e.g., it describes incremental work), and it can significantly benefit from another round of revision. However, I won't object to accepting it if my co-reviewers champion it.

**Missing References:**

Jennifer Hu, Jon Gauthier, Peng Qian, Ethan Wilcox, and Roger Levy. 2020. A systematic assessment of syntactic generalization in neural language models. In Proceedings of the 58th Annual Meeting of the Association for Computational Linguistics, pages 1725–1744, Online. Association for Computational Linguistics.


**Paper Topic And Main Contributions:**

This work focuses on the generalization ability of language embedding in different syntactic contexts in language models used for natural language processing. While these models have proved successful in capturing the semantic aspects of a text, their ability to adapt to a wide variety of syntactic contexts requires more exploration. To investigate, they developed an evaluation set named LSR, which explores three key syntactic aspects: "Lexical compositionally," "Structural heuristics," and "Relational understanding" among concepts. Their analysis revealed that current text embedding models are not meeting these syntactic understanding challenges effectively. Furthermore, they perform some analysis to identify the factors causing these limitations. In addition, they propose a simple data augmentation method to improve the model's performance.

**Questions For The Authors:**

A. The structural heuristic example in Figure 2-a didn't seem clear to me. Can you clarify that?
B. Can you please clarify the ChatGPT model and date of use?
C. You claimed, "This configuration not only ensures a holistic evaluation but also effectively pinpoints the models’ strengths and weaknesses in diverse settings" in line 166. There are some arguments in the discussion manifesting that. Can you please explain how you find models models’ strengths and weaknesses using the results of your benchmark?
D. Verification was a bit vague to me. Can you please re-explain the data generation verification process? How do you make sure the correct verb is chosen in the Verb part of Lexical Compositionality?

**Reasons To Accept:**

1. The manuscript proposes a robust benchmark featuring a well-reasoned testing hypothesis to appraise the syntactic capabilities of LLMs. Also, their benchmark comes from a variety of sources which make provides a mixture of real-world sentence structures.
2. Their benchmark covers 3 aspects that are vital for syntactic generalization.
3. The authors provide a thorough analysis of the results, strengthening their hypotheses and delivering a comprehensive understanding of the results.



**Reasons To Reject:**

1. Although the related work section discusses a range of studies, it overlooks some recent works on syntax generalization, including (https://aclanthology.org/2021.acl-long.289v2.pdf), which delves into systematic generalization with large language models by providing a benchmark. This work would be more compelling by direct comparison of their benchmark with others.
2. The paper does acknowledge the beneficial impact of data augmentation and proposes a simple method to bolster syntactic generalization. However, it falls short by not comparing this method with recent techniques, such as (https://arxiv.org/pdf/2112.07610.pdf).
3. While the authors claim to explore why prior evaluations failed to detect model ineffectiveness, they do not convincingly substantiate this with evidence or clearly elucidate how their benchmark differs from earlier works. Notable missing includes related works such as (https://aclanthology.org/2020.acl-main.158.pdf).
4. The study reports an ~83% agreement rate between human and AI annotators, a figure which may not be sufficiently robust for evaluating large language models. This discrepancy could potentially compromise future evaluations of larger language models, undermining the contributions of this work to the field.
5. Despite utilizing ChatGPT for data annotation, the paper does not specify the version and the date of the model employed for data generation, which may limit the study's applicability.
6. Line 336 makes a claim about models performance on the foundational datasets, but it doesn't provides any results or citation for the claim,

**Reproducibility:**

4: Could mostly reproduce the results, but there may be some variation because of sample variance or minor variations in their interpretation of the protocol or method.

**Reviewer Confidence:**

4: Quite sure. I tried to check the important points carefully. It's unlikely, though conceivable, that I missed something that should affect my ratings.

---

> ### Author Rebuttal · Authors · 2023-08-29
>
> Thank you for your valuable insights and constructive feedback.
>
> > **Q1:** Although the related work section discusses a range of studies, it overlooks some recent works on syntax generalization, including (https://aclanthology.org/2021.acl-long.289v2.pdf), which delves into systematic generalization with large language models by providing a benchmark. This work would be more compelling by direct comparison of their benchmark with others.
> >
>
> **A1:** There might be some misunderstandings.  Our primary goal is to emphasize the limitations of existing **text embedding models instead of large language models** in comprehending syntax. We achieve this by introducing our LSR benchmark. It's worth noting that the benchmarks proposed in the referenced study are designed to **assess large language modeling rather than text embeddings**. We appreciate the reviewer for bringing up the related work on syntax probing, and we intend to address this work in our revised version.
>
> > **Q2:** The paper does acknowledge the beneficial impact of data augmentation and proposes a simple method to bolster syntactic generalization. However, it falls short by not comparing this method with recent techniques, such as ([https://arxiv.org/pdf/2112.07610.pdf](https://arxiv.org/pdf/2112.07610.pdf)).
> >
>
> **A2:** Thank you for mentioning the related study. Our work focused on developing a new benchmark to test the syntax understanding of text embedding models. We have also shown that automatic data augmentation using LLMs effectively addresses these models' limitations. The cited study deals with data augmentation through example recombination to boost the compositional generalization of semantic parsers. Their method depends on the QCFG rules and may not work for all syntactic generalizations. We will discuss and address this work in our upcoming version.
>
> > **Q3:** While the authors claim to explore why prior evaluations failed to detect model ineffectiveness, they do not convincingly substantiate this with evidence or clearly elucidate how their benchmark differs from earlier works. Notable missing includes related works such as ([https://aclanthology.org/2020.acl-main.158.pdf](https://aclanthology.org/2020.acl-main.158.pdf)).
> >
>
> **A3:**  Our primary goal is to emphasize the limitations of existing **text embedding models**  in comprehending syntax. To the best of our knowledge, our work represents the first attempt to construct such a benchmark.  The referenced study is designed to **assess large language modeling rather than text embeddings**. We appreciate the reviewer for bringing up the related work on syntax probing, and we will discuss this work in our revised version.
>
> > **Q4:** The study reports an ~83% agreement rate between human and AI annotators, a figure which may not be sufficiently robust for evaluating large language models. This discrepancy could potentially compromise future evaluations of larger language models, undermining the contributions of this work to the field.
> >
>
> **A4:** Thank you for your comments. It's important to acknowledge that established human annotators frequently demonstrate inter-annotator agreement rates within the 80-90% range, which can vary based on task complexity and the subjectivity inherent in interpreting language. In this context, achieving an agreement rate of around 83% aligns with the inherent diversity in human annotation and establishes a practical benchmark for AI performance.
>
> Furthermore, we have taken steps to ensure the reliability of annotation and generation through human validation. As outlined in section 2.5, we have undertaken a quality validation process using human evaluators. Randomly selected pairs of sentences are presented to human reviewers who assess and verify their reliability. Any sentences that are inaccurately generated will be corrected as needed. The following table presents the statistical breakdown of the revision rate for these corrected sentences:
>
> | Dataset |  | STS |  |  |  | Twitter |  |  |  | CQADupStack |  |
> | --- | --- | --- | --- | --- | --- | --- | --- | --- | --- | --- | --- |
> |  | Samples | Revised samples | Revise rate |  | Samples | Revised samples | Revise rate |  | Samples | Revised samples | Revise rate |
> | Lexical Compositionality | 600 | 28 | 4.67% |  | 600 | 21 | 3.50% |  | 600 | 27 | 4.50% |
> | Structural Heuristics | 400 | 3 | 0.75% |  | 400 | 5 | 1.25% |  | 400 | 9 | 2.25% |
> | Relational Understanding | 400 | 4 | 1.00% |  | 400 | 8 | 2.00% |  | 400 | 6 | 1.50% |
>
> The data indicates a notably low revision rate within the sampled set, showcasing the high quality of sentences generated by ChatGPT. We will incorporate this updated table in the revised version of our work.
>
> > **Q5:**  Despite utilizing ChatGPT for data annotation, the paper does not specify the version and the date of the model employed for data generation, which may limit the study's applicability.
> >
>
> **A5:** The api version of ChatGPT is gpt-3.5-turbo-0301. We will update this information in the revised version.
>
> > **Q6:** Line 336 makes a claim about models performance on the foundational datasets, but it doesn't provides any results or citation for the claim,
> >
>
> **A6:** We have mentioned this claim and referenced the related study in the introduction (see line 41). We'll also add the citation to line 336.
>
> > **Q7:** The structural heuristic example in Figure 2-a didn't seem clear to me. Can you clarify that?
> >
>
> **A7:** In Figure 2-a, the sentence "Her hair is being styled by the girl" serves as a passive transformation of "A girl is styling her hair." This transformation involves rearranging elements: the original object becomes the new subject, and the past participle of the verb is preceded by a form of "to be." This shift in structure is a fundamental attribute of passive voice constructions in the English language. We will make this clear in the revised version.
>
> > **Q8:** Can you please clarify the ChatGPT model and date of use?
> >
>
> **A8:** The api version of ChatGPT is gpt-3.5-turbo-0301. We will update this information in the revised version.
>
> > **Q9:** You claimed, "This configuration not only ensures a holistic evaluation but also effectively pinpoints the models’ strengths and weaknesses in diverse settings" in line 166. There are some arguments in the discussion manifesting that. Can you please explain how you find models models’ strengths and weaknesses using the results of your benchmark?
> >
>
> **A9:** The text embedding models examined in our study, such as sentence-T5, SBERT, Instructor, OpenAI-ada, and SimCSE, have demonstrated impressive performance on conventional text embedding assessments (the original datasets of our LSR benchmark). This underscores these models' robust capacity to be applied across diverse domains. You can find comprehensive results in [1], and this point is also touched upon in our paper on line 41. Due to space constraints, we have not included the performance outcomes of the original benchmarks in this version. However, we intend to integrate this information in the revised version.
>
> However, all of these models exhibit weak performance with low correlations on the LSR benchmark. This indicates that existing text embedding models have not been adequately fine-tuned to effectively address the challenge of syntactic comprehension.
>
> Furthermore, among the aforementioned text embedding models, sentence-T5 consistently outperforms the others in our LSR benchmark, which incorporates syntactic variations not present in the initial datasets. This leads us to posit that the training data utilized by sentence-T5, derived from web-based question-answering sources, likely encompasses a more diverse array of syntactic structures in comparison to other types of data.
>
> > **Q10:** Verification was a bit vague to me. Can you please re-explain the data generation verification process? How do you make sure the correct verb is chosen in the Verb part of Lexical Compositionality?
> >
>
> **A10:** Thank you for the comments. As outlined in section 2.5, we have undertaken a quality validation process using human evaluators. Randomly selected pairs of sentences are presented to human reviewers who assess and verify their reliability. Any sentences that are inaccurately generated will be corrected as needed. The following table presents the statistical breakdown of the revision rate for these corrected sentences:
>
> | Dataset |  | STS |  |  |  | Twitter |  |  |  | CQADupStack |  |
> | --- | --- | --- | --- | --- | --- | --- | --- | --- | --- | --- | --- |
> |  | Samples | Revised samples | Revise rate |  | Samples | Revised samples | Revise rate |  | Samples | Revised samples | Revise rate |
> | Lexical Compositionality | 600 | 28 | 4.67% |  | 600 | 21 | 3.50% |  | 600 | 27 | 4.50% |
> | Structural Heuristics | 400 | 3 | 0.75% |  | 400 | 5 | 1.25% |  | 400 | 9 | 2.25% |
> | Relational Understanding | 400 | 4 | 1.00% |  | 400 | 8 | 2.00% |  | 400 | 6 | 1.50% |
>
> The data indicates a notably low revision rate within the sampled set, showcasing the high quality of sentences generated by ChatGPT. We will incorporate this updated table in the revised version of our work.
>
> [1] Niklas Muennighoff, Nouamane Tazi, Loic Magne, and Nils Reimers. Mteb: Massive text embedding benchmark. In EACL 2023

---

### Official Review · Reviewer_H1cQ · 2023-08-05

**Soundness:** 3

**Excitement:**

3: Ambivalent: It has merits (e.g., it reports state-of-the-art results, the idea is nice), but there are key weaknesses (e.g., it describes incremental work), and it can significantly benefit from another round of revision. However, I won't object to accepting it if my co-reviewers champion it.

**Paper Topic And Main Contributions:**

This paper argues that the existing text embedding models are inadequate for syntax understanding. To this end, the authors introduce a new benchmark LSR for assessing a model's understanding of syntax. LSR  is designed from three syntactic dimensions: Lexical compositionality, Structural heuristics and Relational understanding among concepts. They also propose a data augmentation strategy using examples tailored for  syntactic understanding, to improve model's performance on LSR benchmark.

**Questions For The Authors:**

A. In section 2.2, is it possible that the words being inserted or replaced are unrelated to the original semantics of the sentence at all? Take an example, is it possible the word "cat" in sentence "The cat jumped over the fence" is replaced by "car" rather than "dog"? would it reduce the reliability of the benchmark?
B. Will the corpus be public？

**Reasons To Accept:**

1. The idea of this paper is reasonable and intersting.
2. They develope a new benchmark for syntac understanding.
3. They propose a simple but effective strategy to improve model's understanding of syntax.

**Reasons To Reject:**

Thought the authors reported the correlation scores between the ChatGPT annotations and human annotations, the reliability of the generated sentences is expected to improve further.

**Reproducibility:**

3: Could reproduce the results with some difficulty. The settings of parameters are underspecified or subjectively determined; the training/evaluation data are not widely available.

**Reviewer Confidence:**

4: Quite sure. I tried to check the important points carefully. It's unlikely, though conceivable, that I missed something that should affect my ratings.

---

> ### Author Rebuttal · Authors · 2023-08-29
>
> Thank you for your valuable insights and constructive feedback.
>
> > **Q1:** Thought the authors reported the correlation scores between the ChatGPT annotations and human annotations, the reliability of the generated sentences is expected to improve further.
> >
>
> **A1:** Thank you for the suggestions. As outlined in section 2.5, we have undertaken a quality validation process using human evaluators. Randomly selected pairs of sentences are presented to human reviewers who assess and verify their reliability. Any sentences that are inaccurately generated will be corrected as needed. The following table presents the statistical breakdown of the revision rate for these corrected sentences:
>
> | Dataset |  | STS |  |  |  | Twitter |  |  |  | CQADupStack |  |
> | --- | --- | --- | --- | --- | --- | --- | --- | --- | --- | --- | --- |
> |  | Samples | Revised samples | Revise rate |  | Samples | Revised samples | Revise rate |  | Samples | Revised samples | Revise rate |
> | Lexical Compositionality | 600 | 28 | 4.67% |  | 600 | 21 | 3.50% |  | 600 | 27 | 4.50% |
> | Structural Heuristics | 400 | 3 | 0.75% |  | 400 | 5 | 1.25% |  | 400 | 9 | 2.25% |
> | Relational Understanding | 400 | 4 | 1.00% |  | 400 | 8 | 2.00% |  | 400 | 6 | 1.50% |
>
> The data indicates a notably low revision rate within the sampled set, showcasing the high quality of sentences generated by ChatGPT. We will incorporate this updated table in the revised version of our work.
>
> > **Q2:** In section 2.2, is it possible that the words being inserted or replaced are unrelated to the original semantics of the sentence at all? Take an example, is it possible the word "cat" in sentence "The cat jumped over the fence" is replaced by "car" rather than "dog"? would it reduce the reliability of the benchmark?
> >
>
> **A2:** There is a procedure of scoring the two sentences after the sentence generation, and if “cat” is replaced with “car”, according to similarity scores with explanations and English examples from [1], the generated sentence will be scored low similarity with the anchor sentence. As explained in A1, human reviewers will also carefully examine and address any quality concerns. The statistics highlight that the sentences generated by ChatGPT exhibit a remarkably high level of quality and coherence.
>
> > **Q3:** Will the corpus be public？
> >
>
> **A3:** Yes, we indicated that LSR benchmark and code will be released.(line 103)
>
> [1] Eneko Agirre, Daniel Cer, Mona Diab, Aitor Gonzalez-Agirre, and Weiwei Guo. 2013. *SEM 2013 shared task: Semantic Textual Similarity. In Proceedings of *SEM 2013.

---

### Official Review · Reviewer_QKSJ · 2023-08-06

**Soundness:** 3

**Excitement:**

3: Ambivalent: It has merits (e.g., it reports state-of-the-art results, the idea is nice), but there are key weaknesses (e.g., it describes incremental work), and it can significantly benefit from another round of revision. However, I won't object to accepting it if my co-reviewers champion it.

**Paper Topic And Main Contributions:**

This paper studies how well sentence embedding models understand several syntactic aspects including lexical compositionality, structural heuristics, and relational understanding. To this end, this work proposes a new benchmark for evaluating syntactic understanding of sentence embedding models by applying syntactic transformations to existing datasets with ChatGPT. Results on this benchmark suggest that existing sentence embedding models do not perform well in terms of the three syntactic aspects, and can be improved with targeted data augmentation.

**Questions For The Authors:**

A. Can you provide more details about how you retrain the SBERT model in section 4?

**Reasons To Accept:**

1. The paper is well-written and easy to follow in general.
2. The proposed benchmark identifies the limitations of syntactic understanding of existing sentence embedding models, and provides a valuable resource for improving sentence embeddings in terms of some syntactic aspects that are overlooked by previous work.

**Reasons To Reject:**

1. It is unclear to me why insertion and replacement of a noun/verb/adjective (or lexical compositionality as defined by the authors) can be considered syntactic changes. These operations do not change the sentence structure, but rather add or alter semantic content. I think concept order manipulation is in fact more related to lexical compositionality, as it studies how the order of subject and object affects the sentence-level semantics.
2. It is important to include the performance on the original datasets, especially when the models compared are trained on different datasets/domains. For example, the authors claim that sentence-t5, which is trained on web-based QA datasets, is more robust than other baselines. However, it is unclear whether such improvement is a result of better syntactic understanding, or simply due to sentence-t5 being trained on the same domain as the benchmark dataset.
3. In section 4 the authors retrain the sentence embedding model on syntactically-perturbed inputs with annotations unaltered, and claim that the model lacks syntactic understanding since the performance on the standard STS test set is on par with the original score. However, such settings somewhat encourage the model to ignore syntactic structures given unaltered labels, especially if the model is further finetuned from a trained SBERT model.

**Reproducibility:**

4: Could mostly reproduce the results, but there may be some variation because of sample variance or minor variations in their interpretation of the protocol or method.

**Reviewer Confidence:**

4: Quite sure. I tried to check the important points carefully. It's unlikely, though conceivable, that I missed something that should affect my ratings.

---

> ### Author Rebuttal · Authors · 2023-08-29
>
> Thank you for your valuable insights and constructive feedback.
>
> > **Q1:** It is unclear to me why insertion and replacement of a noun/verb/adjective (or lexical compositionality as defined by the authors) can be considered syntactic changes. These operations do not change the sentence structure, but rather add or alter semantic content. I think concept order manipulation is in fact more related to lexical compositionality, as it studies how the order of subject and object affects the sentence-level semantics.
> >
>
> **A1:** Insertion and replacement of parts of speech (e.g., nouns, verbs, adjectives) present challenges for syntactic generalization. Such concepts about syntax have been discussed in related papers [1,2] or textbook [3], which is the insight we borrowed from.
>
> As mentioned in chapter 2 of [1] and [3],  words in sentences often have hierarchical relationships, represented by tree structures in formal syntactic theories.  Inserting or replacing words can disrupt or modify these hierarchies.  For example, given a sentence: "He shot the man with the gun."  and the replacement: "He shot the man with the camera." In this example, replacing "gun" with "camera" creates syntactic ambiguity. In the original sentence, "with the gun" likely indicates the instrument used for shooting, while in the revised sentence, it can be interpreted as the man having a camera. This change introduces different parsing options and affects syntactic interpretation.
>
> Moreover, parts of sentences can work together as one group, while changing words can break up these groups or make new ones. For instance, given a sentence: "I enjoy coffee.” and the  insertion example: “I enjoy making coffee.”. The  verb "enjoy" originally took a simple direct object, "coffee." Now, it's taking a gerund phrase, "making coffee," which has its own internal structure. Such insertion can alter the composition and boundaries of syntactic constituents. These constituents are pivotal in parsing and understanding sentence structure, as they represent the building blocks of sentences and play a crucial role in comprehending how words are grouped together.
>
> > **Q2:** It is important to include the performance on the original datasets, especially when the models compared are trained on different datasets/domains. For example, the authors claim that sentence-t5, which is trained on web-based QA datasets, is more robust than other baselines. However, it is unclear whether such improvement is a result of better syntactic understanding, or simply due to sentence-t5 being trained on the same domain as the benchmark dataset.
> >
>
> **A2:** The text embedding models studied in our research, such as Sentence-T5, SBERT, Instructor, OpenAI-ada, and SimCSE, have shown impressive performance on standard text embedding tests (the original datasets of our LSR benchmark). This demonstrates these models' strong ability to apply across different domains. You can find the detailed results in [4], and we also touch upon this point in our paper on line 41. Due to space limitations, we haven't included the performance results of the original benchmarks in this version. We will incorporate this information in the updated version.
>
> Among these text embedding models, Sentence-T5 tends to outperform the others in our LSR benchmark, which includes syntactic variations not present in the original datasets. Hence, we propose that the training data used by Sentence-T5, which is sourced from web-based question-answering, might encompass a more diverse range of syntactic structures compared to other data types.
>
> We use tree kernel [5] as a measure of sentence structure diversity. The central idea of tree kernel is to count the number of common subtrees between two constituency pars. We compare the syntactic diversity of two corpora: Web_QA (233k sentences) and NLI (275k sentences). We use StanfordCoreNLP to obtain constituency parse trees of sentences. Then, we randomly select 1,000 parse trees and use the tree kernel to calculate their similarity. The results are shown below. The lower the score, the more diverse the sentence structures are.
>
> | Corpus | Tree Kernel Similarity |
> | --- | --- |
> | Web_QA | 0.064 |
> | NLI | 0.072 |
>
> As we can see, sentences in Web_QA have lower tree kernel similarity than those in NLI, indicating that Web_QA has more diverse sentence patterns and structures.
>
> Since prevailing text embedding models like SBERT, SimCSE, etc. are mainly trained on NLI, we argue that Sentence-T5, which is trained on Web_QA, can capture more syntactic nuances of sentences. This additional experiment supports our hypothesis.
>
> > **Q3:** In section 4 the authors retrain the sentence embedding model on syntactically-perturbed inputs with annotations unaltered, and claim that the model lacks syntactic understanding since the performance on the standard STS test set is on par with the original score. However, such settings somewhat encourage the model to ignore syntactic structures given unaltered labels, especially if the model is further finetuned from a trained SBERT model.
> >
>
> **A3:** Thanks for your comments. There might be some misunderstandings. In section 4, we aim to demonstrate that traditional text embedding evaluation paradigms lack sensitivity to syntactic nuances. To demonstrate this, we retrained the SBERT model  from the scratch with syntax-perturbed data while keeping the annotations unaltered. The results in Figure 3 show that the retrained SBERT model continued to yield good performance on the traditional STS evaluation set. **Specifically, the correlation score following syntax perturbation stood at 85.2%, nearly on par with the original score of 86.2%**. Such empirical results suggest that models can achieve high performance on traditional text embedding evaluation benchmarks without proper syntactic understanding. These findings prompt a more rigorous and discerning evaluation paradigm that factors in both semantic and syntactic elements, paving the way for more accurate assessments of text embedding models.
>
> > **Q4:** Can you provide more details about how you retrain the SBERT model in section 4?
> >
>
> **A4:** We retrain the SBERT model from the scratch using the same training objective and datasets. The only difference is the incorporation of syntax-perturbed data into the training dataset. STS training dataset consists of Sentence1, Sentence 2, and its semantic similarity score is marked by humans. We denote it as **[S1, S2, score]**.
>
> In Section 4.1, we fix Sentence 1 and do the Order Perturbation and Relational Perturbation on each Sentence 2. Each sentence pair is changed into **[S1, Perturbed S2, score]**. So the model retrained can not learn proper syntactics at all, they can just learn how far the distance is in text embedding space between contents in two sentences.
>
> In Section 4.2, for the simple data augmentation, we fix Sentence 1, do Lexical compositionality, Structural heuristics, and Relational understanding on each Sentence 1 and ask ChatGPT(gpt-3.5-turbo-0301) to score the semantic similarity between the fixed Sentence 1 and its perturbed one based on similarity scores with explanations and English examples from [6]. We manually verified the generation reliability and annotation rationality. Each sentence pair can be written as **[S1, Perturbed S1, re-score]**. The model retrained can learn proper syntactics variants with correct semantic scores.
>
> Through enhancing the STS dataset, we finally obtain 10k training examples, 1.5k dev examples and 1.3k test samples.
>
> We choose microsoft/mpnet-base as our raw model which is also the basic model of the best SBERT text embedding model SBERT-all-mpnet-base-v2. We follow Sentence-BERT training settings [7], use the regression objective function, a batch-size of 16, 4 training epochs, Adam optimizer with learning rate 2e−5, and a linear learning rate warm-up over 10% of the training data. Our default pooling strategy is MEAN. We save the best parameters according to the dev set at the end of each epoch. At evaluation time, we compute the cosine-similarity between the sentence embeddings. All systems are trained with 10 random seeds to counter variances. We will clarify these settings in our revised manuscript.
>
> [1] Barbara H. Partee. 1995. Lexical semantics and compositionality.
>
> [2] Edward Gibson. 1998. Linguistic complexity: locality of syntactic dependencies. Cognition.
>
> [3] Andrew Carnie. 2012. Syntax: A generative introduction, volume 18. John Wiley & Sons.
>
> [4] Niklas Muennighoff, Nouamane Tazi, Loic Magne, and Nils Reimers. Mteb: Massive text embedding benchmark. In EACL 2023
>
> [5] Collins, M., & Du y, N. (2002). New ranking algorithms for parsing and tagging: Kernels over discrete structures, and the voted perceptron. In ACL
>
> [6] Eneko Agirre, Daniel Cer, Mona Diab, Aitor Gonzalez-Agirre, and Weiwei Guo. 2013. *SEM 2013 shared task: Semantic Textual Similarity. In Proceedings of *SEM 2013.
>
> [7] Reimers N, Gurevych I. Sentence-BERT: Sentence Embeddings using Siamese BERT-Networks[C]//Proceedings of the 2019 Conference on Empirical Methods in Natural Language Processing and the 9th International Joint Conference on Natural Language Processing (EMNLP-IJCNLP). Association for Computational Linguistics, 2019.

---

### Meta-Review · Area_Chair_LafB · 2023-09-09

**Recommendation:** 4

**Metareview:**

The paper presents a novel dataset to evaluate ability of sentence embedding models (such as SentenceBERT) to distinguish semantic differences expressed using syntactical means: e.g. A dog chase a cat -> A cat chase a dog.
The paper uses a semi-automatic process to expand an existing evaluation dataset and then benchmarks existing sentence models, showing their limitations.
All three reviewers agree in their general assessment of the paper. Authors made significant efforts to further clarify their task during rebuttal period. Parts of their response should be included into the final version, especially the table with human assessment.

---

### Decision · Program_Chairs · 2023-10-07

**Decision:**

Accept-Findings

**Comment:**

The paper presents a novel dataset to evaluate ability of sentence embedding models (such as SentenceBERT) to distinguish semantic differences expressed using syntactical means: e.g. A dog chase a cat -> A cat chase a dog.
The paper uses a semi-automatic process to expand an existing evaluation dataset and then benchmarks existing sentence models, showing their limitations.
All three reviewers agree in their general assessment of the paper. Authors made significant efforts to further clarify their task during rebuttal period. Parts of their response should be included into the final version, especially the table with human assessment.